# Regional Economic Impacts from Timber Check Dam Construction—A Comparison with Concrete Check Dam Construction, Part II: The Question of Premium Vouchers

**Tomohumi Huzita** [1,]*[ ] and **Chihiro Kayo** [2]

1   United Graduate School of Agricultural Science, Tokyo University of Agriculture and Technology, 3-5-8 Saiwai-cho, Fuchu, Tokyo 183-8509, Japan

2   Institute of Agriculture, Tokyo University of Agriculture and Technology, 3-5-8 Saiwai-cho, Fuchu, Tokyo 183-8509, Japan; kayoc@cc.tuat.ac.jp

*   Correspondence: s195462r@st.me.tuat.ac.jp

**Abstract:** The regional economic impacts of the construction of timber and concrete check dams in Akita prefecture, Japan, were compared. When functions are appropriately unified, a concrete check dam is cheaper to build than a timber one; the difference in construction costs can be used by the government to issue premium vouchers as a regional promotion measure. To evaluate the effect of this, we extended the latest input–output table for Akita Prefecture to include timber and concrete check dam construction sectors. Concrete check dam construction with premium-voucher issuance had a total effect of 46,064,000 yen (economic impact ratio: 1.49; increased employment: 4.68 people). It also had an indirect economic impact on the plywood sector, since plywood was used as formwork, and issuing premium vouchers had a ripple effect on many industries in the region. However, timber check-dam construction had a total effect of 57,706,000 yen (economic impact ratio: 1.86; increased employment: 5.11 people) and a larger effect on the timber, logs, and silviculture sectors. Therefore, despite higher initial costs, timber check dam construction has the greater economic and employment impact on a region through its large ripple effects on the forestry and wood industries.

**Keywords:** economic impact; input–output table; input–output analysis; timber check dam; wood industries; regional economy; concrete check dam; premium voucher; Japan; Akita Prefecture

## 1. Introduction

Recently, the frequency and intensity of heavy rainfall has increased in several parts of the world [1]. In areas such as Japan, where the terrain is frequently steep and prone to landslides, typhoons and heavy rains have had calamitous effects in recent years [2]. In 2019, storm damage to forests was massive, costing approximately 80.5 billion yen [3]. Given both the obvious need for disaster prevention and the multifunctional roles played by forests in Japan [3], preservation of wooded areas from storm-induced flooding and erosion is a necessity. One important type of structure meant to achieve this is the check dam, which reduces erosion by mountain streams and helps maintain forests facing them.

Sustainable forest management is being promoted throughout the world [4–6]. In Japan, sustainable forest management involves logging and using trees from planted forests when they reach a suitable age, and replanting trees to ensure preservation of the next generation. Timber (as opposed to concrete) check dams (henceforth, timber dams) have attracted much attention for use in sustainable forestry [7,8]. Besides its direct contribution to the maintenance and improvement of forest function, the use of timber for the check dam contributes to the revitalization of the regional economy, because it involves a variety of industries, the forestry and wood industries among them. Sustainable forest management must be socially beneficial to those who depend on forestry for their livelihood [9]; it is therefore important to get a quantitative handle on the impact of timber dam construction on the regional economy.

Previous research has examined the functions and advantages of check dams, including soil conservation and torrent control [10–14]. Regarding the use of wood for the construction of check dams, certain studies in the literature have evaluated life-cycle $CO_2$ emissions [15,16], estimated the potential for wood use and reduction in greenhouse gas emission [17], and calculated the structural stability [18]. However, with the exception of our previous study [19], no study has investigated the economic impact of timber dams, although a prior research has evaluated the economic impact of using timber as building material [20].

Our previous study was the first attempt at quantifying the regional economic impact of constructing timber dams, viewed as a public works project [19]. It revealed that the use of regionally produced timber in the construction of a timber dam had a positive impact on the regional economy, and its economic impact was greater than that of the construction of a concrete check dam (henceforth, concrete dam). However, there were significant flaws in the previous work. In assessing the contribution of timber dam construction to the regional economy, it is important to compare it with the construction of general non-timber (e.g., concrete) dams. The previous study [19] therefore compared the regional economic impacts of timber and concrete dams, using the amount of sediment runoff prevented as a common functional unit for the comparison. However, the construction cost is different between the two: the concrete dam is cheaper. Therefore, building it rather than a timber dam creates a surplus that can be used for other regional promotion measures such as premium voucher issuance, making the concrete dam appear even more advantageous. It is thus necessary to unify both the functions and costs including the use of the surplus in order to examine the regional economic impact of the timber dam in comparison with the concrete dam. However, our previous study [19] and existing literature [20] did not consider these necessary aspects.

Therefore, in this work, we improve upon the previous study [19] by taking both vouchers and their regional economic impacts into account. The input–output table, the analysis tool used in the previous study, has been updated for this study. By subdividing the forestry and lumber and wood products sectors, we have improved it so that more detailed economic impacts can be evaluated. We find that, despite its higher initial cost, the timber dam has a better regional economic impact than the concrete dam, even when the government issues premium vouchers to regional residents to encourage consumption within a region affected by concrete dam construction. In this paper, we propose a versatile method that can also be applied to the comparison of regional economic impacts of structures and public works other than check dams. Further, this study will help policy makers to consider the impact of the use of timber in public works on the regional economy.

## 2. Methods

### 2.1. Subject Region and Subject Check Dams

The subject region, as in our previous study [19], was Akita Prefecture in Japan. The subject timber dam (Figure 1a) (length 30.6 m; height 3.0 m; lumber consumption 241.8 m$^3$; amount of sediment runoff prevented 602 m$^3$) was built in Kitaakita City, Akita Prefecture, in FY2013. The tree species used in the timber dam is Japanese cedar (*Cryptomeria japonica* D. Don) lumber from Akita Prefecture. Figure 2 shows the front and sectional views of the timber dam. We compare it with a concrete dam (Figure 1b) (length 20.0 m; height 6.0 m; lumber consumption 10.5 m$^3$; concrete consumption 237.5 m$^3$; amount of sediment runoff prevented 320 m$^3$), which was built in Yurihonjo City, Akita Prefecture, in FY2018. Both dams were ordered by the Akita Prefecture as public works.

### 2.2. About Economic Impacts

As in the previous paper [19], we evaluated the production value induced, the gross-value-added induced, the employee income induced, and the employment effect for the direct, indirect, secondary, and total effects with respect to the economic impact.

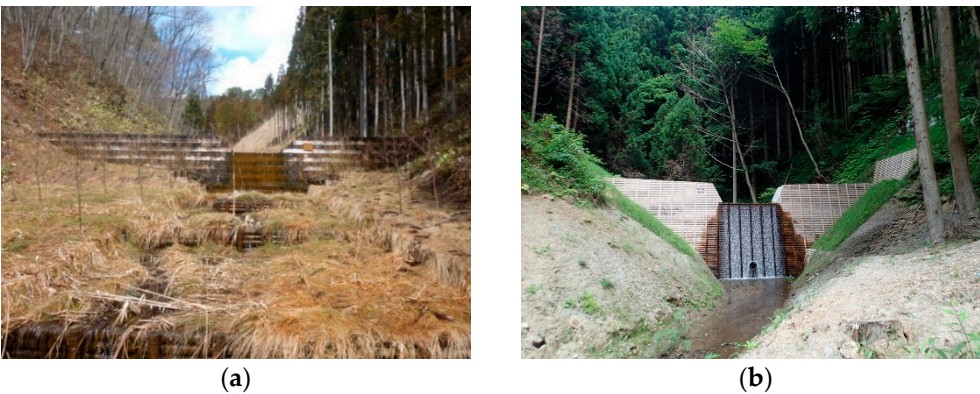

**Figure 1.** (**a**) A timber dam; (**b**) a concrete dam [19].

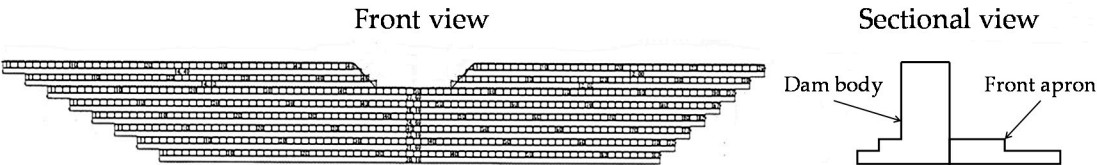

**Figure 2.** Front and sectional views of a timber dam [19].

An input–output table was used to calculate the economic impact. The procedure was as follows: first, new construction sectors for timber and concrete dams were added to the existing input–output table. Then, construction revenue and expenditure (input–output) data were surveyed from check-dam constructors. The data were used to create an extended input–output table with the new sector for the construction of check dams. Based on this extended table, a matrix of input coefficients and an inverse matrix coefficient table were created. The construction cost of the check dams as the final demand increased was considered. For the concrete dam, the final demand increase due to the premium voucher issue was also considered (see Section 2.3 below). In addition, the direct, indirect, and secondary effects were estimated using the above coefficients.

### 2.3. Unification of Functions and Costs of Timber and Concrete Dams

When the functions (amounts of sediment-runoff prevention) of the timber dam and the concrete dam examined in this study are unified, the construction costs are different, and the concrete dam is cheaper. We considered this difference to be available to the government (Akita Prefecture) for other regional promotion measures. Therefore, in order to take account of both functions and costs when comparing timber and concrete dams, this study evaluated the regional economic impact due to induced consumption when the difference in costs is made available to regional residents as government-issued premium vouchers (Figure 3).

Building a timber dam costs 31,001,000 Japanese yen. Building a concrete dam with the same function (amount of sediment runoff prevention) as the timber dam costs 14,595,000 yen. The difference is the prefectural expense of 16,406,000 yen saved by the government if a concrete dam is built instead of a timber one. This money can then be used for regional development, in particular by issuing premium vouchers to regional residents.

For premium vouchers, we referenced the measure [21] that issued premium vouchers in 2015 as a regional promotion measure in Akita City, Akita Prefecture. More specifically, whenever a regional resident purchased a voucher of 1000 yen per sheet, a premium of 200 yen was added, making 1200 yen available for shopping. The premium is the burden borne by the government. The report [21] listed the industries with increased consumption because of the issue of premium vouchers and the extent of increase in the consumption. The amount of new-consumption inducement included the amount borne

by the residents and the amount borne by the government (the premium). We defined the amount borne by the government as the difference in construction costs between the two dams in this study. The increase in the final demand in the amount borne by the government due to the issue of premium vouchers by Akita Prefecture was calculated. First, the new-consumption inducement of each industrial sector was divided by the total new-consumption inducement; this ratio was then multiplied by the difference between the construction costs of the timber and concrete dams.

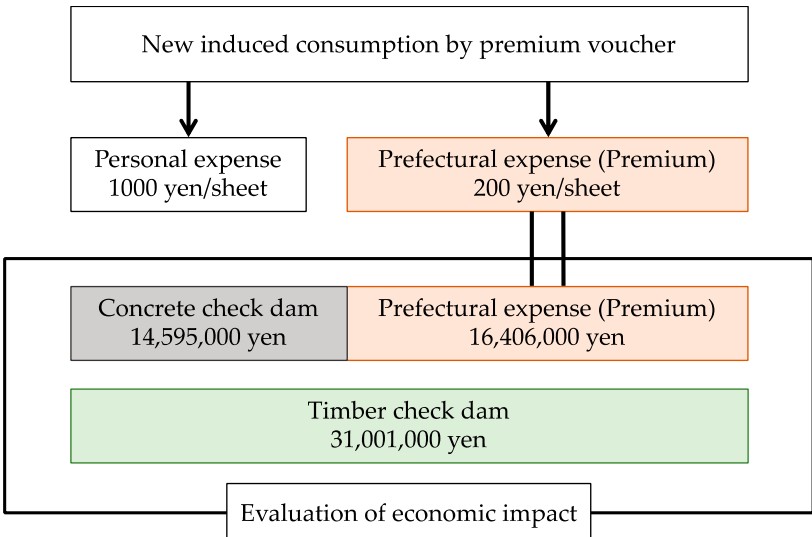

**Figure 3.** A framework for unifying and comparing the functions and costs of timber and concrete dams. It initially costs 16,406,000 yen more to build a timber dam than a concrete one; the difference can be used to add premiums of 200 yen each to government-issued 1000-yen vouchers for regional development.

*2.4. Creation of Extended Input–Output Table*

Figure 4 shows a schematic of the extended input–output table. We used the 2015 Akita Prefecture Input–Output Table (henceforth Akita table) [22], which is the latest input–output table for the prefecture, to evaluate the economic impact. In the Akita table, the industrial sector was divided into 107 sectors. There are only two categories in the forestry and wood industry, the lumber and wood products sector and the forestry sector. However, the wood used for timber dam construction (mainly lumber used as the main material) and the wood used for concrete dam construction (mainly plywood used for the formwork) are different. In addition, the forestry sector in the Akita table includes special forest products (including those related to hunting). The classification is thus too broad to evaluate the economic impact on the forestry and wood industries in detail. Therefore, using the "basic sector table" [23] and "107 sector table" [24], the lumber and wood products sector was divided into timber, plywood and glued laminated timber, wooden chips, and miscellaneous wooden products; while the forestry sector was divided into silviculture, logs, and special forest products (including hunting). In the case of the lumber and wood products sector, the ratio of timber, plywood, and glued laminated timber, wooden chips, and miscellaneous wooden products sectors to the domestic production in the lumber and wood products sector in the basic sector table was calculated, and it was multiplied by the prefectural production in the lumber and wood products sector in the Akita table. The forestry sector in the Akita table was also divided in the same way using the ratio of the silviculture, logs, and special forest products (including hunting) sectors to the forestry sector in the basic sector table. However, in order to unify the number of row and column sectors, the sector of wooden products for construction and the sector of wooden products not elsewhere classified were compounded as a miscellaneous wooden products

sector. The extended input–output table was developed by creating a timber or a concrete dam construction sector and dividing the lumber and wood products sector into timber, plywood and glued laminated timber, wooden chips, and miscellaneous wooden products sectors and the forestry sector into silviculture, logs, and special forest products (including hunting) sectors, for a total of 113 sectors.

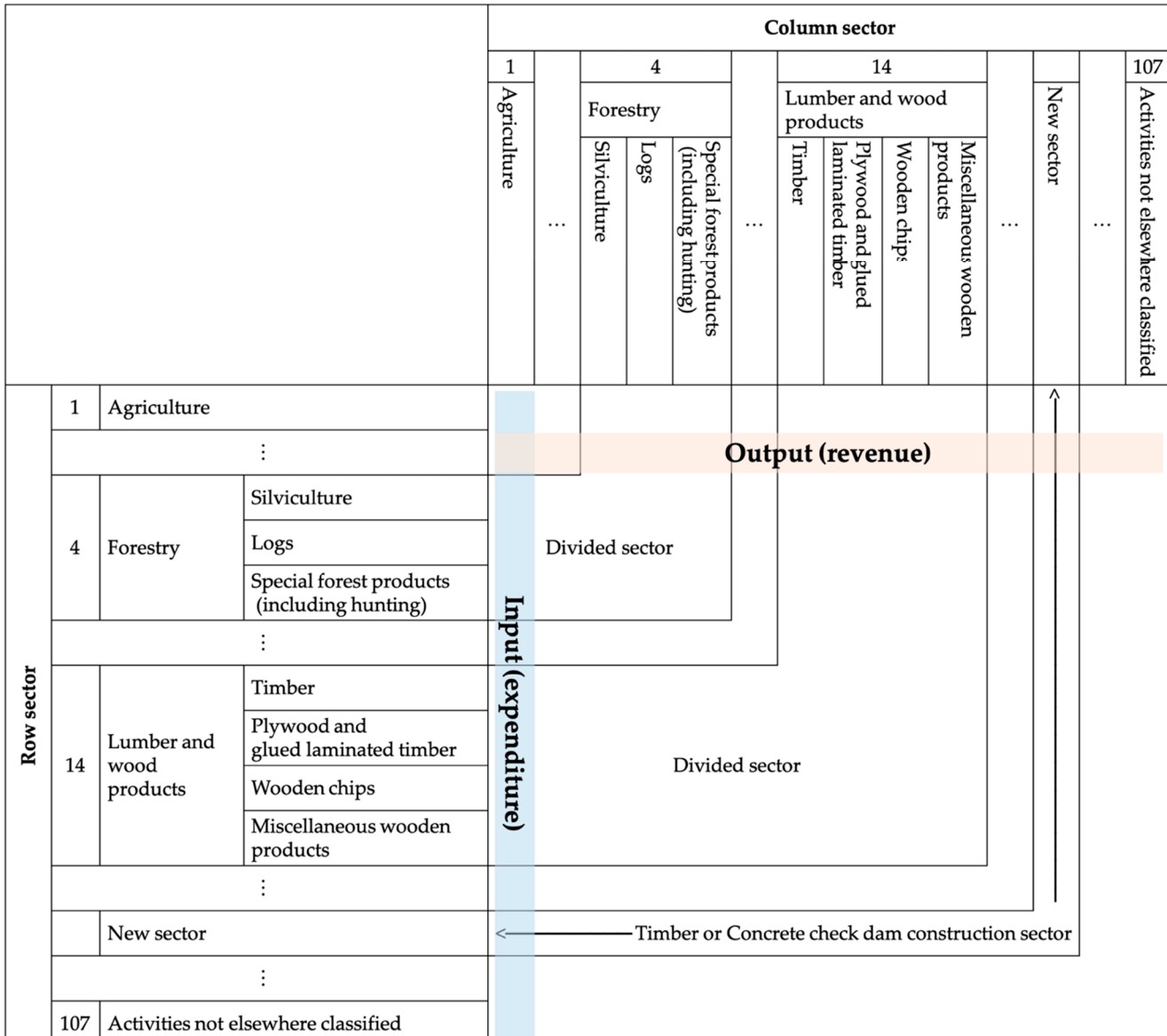

**Figure 4.** Schematic table of the extended input–output table (Extended Akita Table).

### 2.5. Input–Output Data

To create the new construction sectors of timber and concrete dams, we need detailed data of construction revenue and expenditure such as construction, material, and personal costs. In this study, we used the data investigated in the previous paper [19]. The 2015 input–output tables for Japan [25] were used to classify the revenue and expenditure data into the corresponding sectors. The materials and expenses and the corresponding industrial sectors of constructing a timber and a concrete dam investigated in the previous paper are shown in Tables 1 and 2, respectively. Table 3 shows the industrial sectors in which the final demand increase for premium vouchers occurred.

**Table 1.** Timber dam: The materials and expenses and the corresponding industrial sectors [19].

| Materials and Expenses | Sector | |
| --- | --- | --- |
| | Intermediate Sectors | Gross Value Added |
| Lumber | Timber | |
| Lag screw | Metal products for construction and architecture | |
| Steel | Metal products for construction and architecture | |
| Broken rock | Non-metallic ores | |
| Trees (afforestation) | Silviculture | |
| Turfs | Crop cultivation | |
| Machine rental | Goods rental and leasing services | |
| Fuel | Petroleum refinery products | |
| Equipment | General-purpose machinery | |
| Communication | Communication | |
| Construction insurance | Finance and insurance | |
| Bank guarantee charge | Finance and insurance | |
| Miscellaneous expenses | Office supplies | |
| Personal costs | | Wages and salaries |
| Retirement fund | | Wages and salaries |
| Tax and public dues | | Indirect tax |

**Table 2.** Concrete dam: The materials and expenses and the corresponding industrial sectors [19].

| Materials and Expenses | Sector | |
| --- | --- | --- |
| | Intermediate Sectors | Gross Value Added |
| Form plywood | Plywood and glued laminated timber | |
| Freshly mixed concrete | Cement and cement products | |
| Structural steel | Steel products | |
| Fuel | Petroleum-refinery products | |
| Artificial coal | Coal products | |
| Concrete-shell disposal | Waste management service | |
| Insurance | Finance and insurance | |
| Interest | Finance and insurance | |
| Machine rental | Goods rental and leasing services | |
| Turfs | Crop cultivation | |
| Vegetation sandbags | Crop cultivation | |
| Material-storage | Storage-facility service | |
| Blue tarp | Plastic products | |
| Miscellaneous expenses | Commerce | |
| Personal costs | | Wages and salaries |
| Statutory welfare costs | | Wages and salaries |
| Welfare expenses | | Welfare expenses |
| Tax and public dues | | Indirect tax |

The construction and administration years are different in the input–output table (2015), the concrete dam (FY2018), the timber dam (FY2013), and the premium voucher (FY2015). Using construction-cost deflators [26], the construction costs of concrete and timber dams were converted to those in 2015, the publication year of the input–output table. The construction cost of the concrete dam in 2015 was obtained by dividing the deflator in FY2015 by the deflator in FY2018 and multiplying the result by the concrete dam construction cost. The construction cost of the timber dam in FY2013 was converted to the 2015 cost in the same way. However, the premium voucher was not converted, because it was implemented in FY2015.

**Table 3.** Industrial sectors in which final demand is increased by premium voucher.

| Items Purchasable with Premium Voucher | Intermediate Sectors |
| --- | --- |
| Fresh food | Crop cultivation; fishery and foods |
| Processed foods and beverages | Foods and beverages |
| Clothes and bedclothes | Wearing apparel and miscellaneous ready-made textile products |
| Furniture and fixtures | Furniture and fixtures |
| Electric appliances | Household electric appliances |
| Jewelry, bag, and leather products | Tanned leather; leather products; furs and hides |
| Timepieces and glasses | Miscellaneous manufacturing products |
| Drugs | Medicaments |
| Cosmetics | Final chemical products (except medicaments) |
| Kitchenware | Miscellaneous manufacturing products |
| Automobiles | Passenger motor cars |
| Automobile parts | Motor-vehicle parts and accessories |
| Toys and entertainment supplies | Miscellaneous manufacturing products |
| Housing improvements | Building construction |
| Entertainment and leisure | Miscellaneous personal services |
| Eating out | Eating and drinking services |
| Beauty salon | Cleaning, barber shops, beauty shops, and public baths |
| Education | Printing, plate making, and book binding; miscellaneous personal services |
| Other | Activities not elsewhere classified |

### 2.6. Trade Margins and Domestic Freight

The revenue and expenditure data investigated were purchaser's prices including trade margin and domestic freight. Therefore, it was necessary to remove the trade margin and domestic freight from the purchaser's price to obtain the producer's price. The trade margin/domestic freight ratio [27] was used to calculate the producer's price. The calculated producer's prices were assigned to the corresponding column of each check dam sector. The trade margin and domestic freight that were removed were allocated to the commerce and transport sectors (a total of six sectors) of the column of the timber or concrete dam-construction sectors.

### 2.7. Prefecture's Self-Sufficiency Ratios

The subject timber dam was constructed using 100% regionally produced lumber (harvested and processed in Akita Prefecture). Therefore, to set 100% self-sufficiency ratio of the prefecture in the logs and timber sectors of the industrial sector corresponding to timber harvesting and processing, the import coefficients for these sectors were set to 0. The present self-sufficiency ratios in Akita Prefecture are 24% in the timber sector and 62% in the logs sector. Thus, the import coefficients are presently 0.76 and 0.38, respectively.

### 2.8. Evaluation of the Economic Impact

Figure 5 shows the procedure for economic impact evaluation. The construction cost of the timber dam was used as the final demand increase. In the case of the concrete dam, the final demand increase was taken from the construction cost of the concrete dam and the amount borne by the government in issuing premium vouchers.

The matrix of input coefficients (*A*) is a square matrix with (*i*, *j*) element

$$a_{ij} = \frac{x_{ij}}{X_j} \tag{1}$$

where *i* represents the row number, *j* represents the column number, $a_{ij}$ is the input to *sector j* from *sector i* to produce one unit of product; $x_{ij}$, the input to *sector j* from *sector i*; $X_j$, the prefecture's production of *sector j*. The inverse matrix coefficient table (*B*) is calculated from:

$$B = [I - (I - M)A]^{-1} \tag{2}$$

where *I* is the unit matrix; *M* is a diagonal matrix with import coefficients as diagonal elements and zero-valued non-diagonal elements.

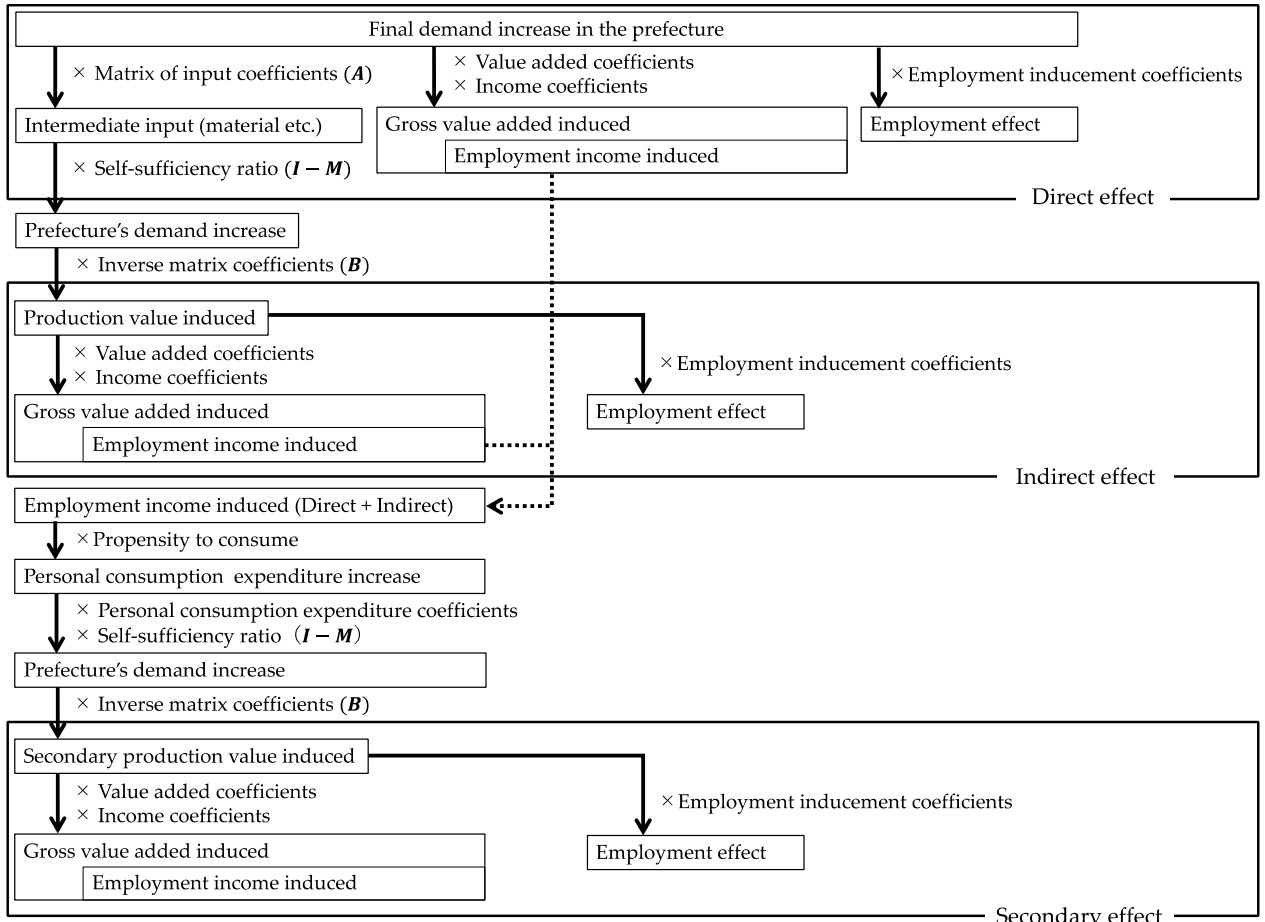

**Figure 5.** Procedure for evaluating economic impact.

The value-added coefficients were defined as the gross value added of each sector divided by the prefecture's production for that sector. The income coefficients were defined as the income of employees in each sector divided by the prefecture's production for that sector. The propensity to consume was defined as the consumption expenditure divided by the real income, using the 2015 family income and expenditure survey [28]. The personal-consumption expenditure coefficients were defined as the consumption expenditure of each sector divided by the total consumption expenditure.

The employment-inducement coefficients were calculated by dividing the number of employees by the production values. However, for the employment-inducement coefficients of the newly created timber and concrete dam construction sectors, the employment-inducement coefficients of the existing public-construction sector were used because reliable data could not be collected. In addition, the number of employees in the timber, plywood and glued laminated timber, wooden chips, and miscellaneous wooden products sectors (all originally composing the lumber and wood products sector), and the silviculture, logs, and special forest products (including hunting) sectors, (all originally in the forestry sector), was defined by the same method as in Section 2.4. In the case of the lumber and wood products sector, the ratio of the number of employees in the timber, plywood and glued laminated timber, wooden chips, and miscellaneous wooden products sectors to the number of employees in the lumber and wood products sector in the national table was multiplied by the number of employees in the lumber and wood products sector in the

Akita table. The number of employees in the forestry sector in Akita table was also divided in the same way using the ratio of silviculture, logs, and special forest products (including hunting) sectors to the forestry sector in the basic sector table.

## 3. Results

### 3.1. The Regional Economic Impact of a Timber Dam and a Concrete Dam by Unified Functions

Tables 4 and 5 show the economic impacts of the construction of a timber and a concrete dam, respectively. The construction cost of the timber dam is 31,001,000 yen (in 2015 yen). It represents the final demand increase (direct effect) of the timber dam construction sector. As a result, the production value induced was 57,706,000 yen; the gross value added was 24,631,000 yen, and the employee income induced was 15,244,000 yen. The employment effects from the direct, indirect, secondary, and total effects were 2.97, 1.58, 0.57, and 5.11 people, respectively. In addition, the economic impact ratio (ratio of total effect to direct effect) was 1.86. The construction cost (the final demand increase) of the concrete dam is 14,595,000 yen (in 2015 yen) when the amount of sediment-runoff prevention is equal to that of the timber dam. As a result, the production value induced was 23,266,000 yen, the gross value added was 10,671,000 yen, and the employee income induced was 7,598,000 yen. The employment effects from the direct, indirect, secondary, and total effects were 1.40, 0.47, 0.28, and 2.15, respectively. In addition, the economic impact ratio was 1.59.

**Table 4.** The economic impact (measured in thousands of yen or in people) of the construction of a timber dam.

| Title | Direct Effect | Indirect Effect | Secondary Effect | Total Effect | Economic Impact Ratio |
|---|---|---|---|---|---|
| Production value induced | 31,001 | 20,159 | 6546 | 57,706 | 1.86 |
| Gross-value-added induced | 10,467 | 10,066 | 4098 | 24,631 | - |
| Employment income induced | 9006 | 4680 | 1558 | 15,244 | - |
| Employment effect | 2.97 | 1.58 | 0.57 | 5.11 | - |

**Table 5.** The economic impact (measured in thousands of yen or in people) of the construction of a concrete dam, after unifying the amount of sediment-runoff prevention.

| Title | Direct Effect | Indirect Effect | Secondary Effect | Total Effect | Economic Impact Ratio |
|---|---|---|---|---|---|
| Production value induced | 14,595 | 5409 | 3262 | 23,266 | 1.59 |
| Gross-value-added induced | 5632 | 2996 | 2043 | 10,671 | - |
| Employment income induced | 5484 | 1338 | 776 | 7598 | - |
| Employment effect | 1.40 | 0.47 | 0.28 | 2.15 | - |

Tables 6 and 7 show the top five industrial sectors for the indirect and secondary effects of the construction of a timber dam and a concrete dam, along with the percent contributions of those sectors to the production value induced. Regarding the timber dam, the top five sectors for the indirect effects were, in descending order, timber, logs, commerce, silviculture, and goods rental and leasing services. There were 32 sectors with a production value of 10,000 yen or more. By contrast, for the concrete dam, the top five sectors for the indirect effects, in descending order, were commerce, cement and cement products, goods rental and leasing services, finance and insurance, and plywood and glued laminated timber. There were 26 sectors with a production value of 10,000 yen or more. In the case of timber dam construction, a large ripple of indirect effects appeared in the sector of timber (which was used as the main material) and in the logs and silviculture sectors of upstream industry. In the case of concrete dam construction, a large ripple of indirect effects appeared in the sector of cement and cement products (again, the main construction material). In addition, since plywood was used for the formwork, there was a large ripple effect in the plywood and glued laminated timber sector as well. The top five sectors for secondary effects were the same for both timber and concrete dam construction: in order, house rent (imputed house rent), commerce, finance and insurance, eating and drinking services, and medical services.

**Table 6.** Indirect and secondary effects (in thousands of yen) of the construction of a timber dam, with percent contribution to production value induced: top five sectors.

| Indirect Effects | | | Secondary Effects | | |
|---|---|---|---|---|---|
| Sector | | | Sector | | |
| Timber | 7002 | 35% | House rent (imputed house rent) | 1273 | 19% |
| Logs | 3649 | 18% | Commerce | 944 | 14% |
| Commerce | 2773 | 14% | Finance and insurance | 567 | 9% |
| Silviculture | 1625 | 8% | Eating and drinking services | 388 | 6% |
| Goods rental and leasing services | 1167 | 6% | Medical service | 280 | 4% |

**Table 7.** Indirect and secondary effects (in thousands of yen) of the construction of a concrete dam, with percent contribution to production value induced: top five sectors.

| Indirect Effects | | | Secondary Effects | | |
|---|---|---|---|---|---|
| Sector | | | Sector | | |
| Commerce | 1257 | 23% | House rent (imputed house rent) | 635 | 19% |
| Cement and cement products | 1256 | 23% | Commerce | 471 | 14% |
| Goods rental and leasing services | 841 | 16% | Finance and insurance | 283 | 9% |
| Finance and insurance | 502 | 9% | Eating and drinking services | 194 | 6% |
| Plywood and glued laminated timber | 319 | 6% | Medical service | 140 | 4% |

### 3.2. The Regional Economic Impact of Timber and Concrete Dams with Unified Functions and Costs

Table 8 shows the regional economic impact of issuing a premium voucher. The economic impacts of the premium voucher from the production value, gross value, and employee income induced were 22,798,000, 10,902,000, and 5,377,000 yen, respectively. The employment effects from the direct, indirect, secondary, and total effects were 2.02, 0.30, 0.20, and 2.52, respectively. In addition, the economic impact ratio was 1.39.

**Table 8.** The economic impact (measured in thousands of yen or in people) of issuing a premium voucher.

| Title | Direct Effect | Indirect Effect | Secondary Effect | Total Effect | Economic Impact Ratio |
|---|---|---|---|---|---|
| Production value induced | 16,406 | 4083 | 2309 | 22,798 | 1.39 |
| Gross-value-added induced | 7614 | 1842 | 1445 | 10,902 | - |
| Employment income induced | 3929 | 899 | 549 | 5377 | - |
| Employment effect | 2.02 | 0.30 | 0.20 | 2.52 | - |

Table 9 shows the top five industrial sectors for the indirect and secondary effects of issuing a premium voucher, along with the percent contributions of those sectors to the production value induced. The top five industrial sectors for the indirect effects of premium voucher issue, in order, were self-transport, commerce, miscellaneous business services, electricity, and finance and insurance. There were 43 sectors with a production value of 10,000 yen or more.

Table 10 shows the economic impact of constructing a concrete dam with unified functions and costs and issuing a premium voucher. The economic impacts of concrete dam construction and premium voucher on the production value, gross value, and employee income induced were 46,064,000, 21,573,000, and 12,976,000 yen, respectively. The employment effects from the direct, indirect, secondary, and total effects were 3.42, 0.78, 0.48, and 4.68 people, respectively. Furthermore, the economic impact ratio was 1.49. Thus, even if the difference in construction costs is used to induce new consumption by issuing premium vouchers, the regional economic impact of constructing a timber dam is greater than that of a concrete dam.

**Table 9.** Indirect and secondary effects (in thousands of yen) of issuing a premium voucher, with percent contribution to production value induced: top five sectors.

| Indirect Effects | | | Secondary Effects | | |
|---|---|---|---|---|---|
| Sector | | | Sector | | |
| Self-transport | 684 | 17% | House rent (imputed house rent) | 449 | 19% |
| Commerce | 535 | 13% | Commerce | 333 | 14% |
| Miscellaneous business services | 384 | 9% | Finance and insurance | 200 | 9% |
| Electricity | 273 | 7% | Eating and drinking services | 137 | 6% |
| Finance and insurance | 194 | 5% | Medical service | 99 | 4% |

**Table 10.** Economic impact (in thousands of yen or in people) of the construction of a concrete dam and issuing a premium voucher.

| Title | Direct Effect | Indirect Effect | Secondary Effect | Total Effect | Economic Impact Ratio |
|---|---|---|---|---|---|
| Production value induced | 31,001 | 9492 | 5571 | 46,064 | 1.49 |
| Gross-value-added induced | 13,246 | 4838 | 3488 | 21,573 | - |
| Employment income induced | 9413 | 2237 | 1326 | 12,976 | - |
| Employment effect | 3.42 | 0.78 | 0.48 | 4.68 | - |

Table 11 shows the top five industrial sectors for the indirect and secondary effects, respectively, of constructing a concrete dam and issuing a premium voucher. The top five sectors for the indirect effects of the construction of a concrete dam and issue of premium voucher, in order, were commerce, cement and cement products, goods rental and leasing services, self-transport, and finance and insurance. There were 55 sectors with a production value of 10,000 yen or more.

**Table 11.** Indirect and secondary effects (in thousands of yen) of the construction of a concrete dam and issuing a premium voucher, with percent contribution to production value induced: top five sectors.

| Indirect Effects | | | Secondary Effects | | |
|---|---|---|---|---|---|
| Sector | | | Sector | | |
| Commerce | 1792 | 19% | House rent (imputed house rent) | 1084 | 19% |
| Cement and cement products | 1280 | 13% | Commerce | 804 | 14% |
| Goods rental and leasing services | 906 | 10% | Finance and insurance | 482 | 9% |
| Self-transport | 876 | 9% | Eating and drinking services | 331 | 6% |
| Finance and insurance | 695 | 7% | Medical service | 239 | 4% |

## 4. Discussion

Table 6 shows that the construction of a timber dam has a large indirect effect on the forestry and wood industries, specifically the timber, logs, and silviculture sectors. The indirect effect on the timber sector is because of the increase in demand for lumber used as the main material in the construction of a timber dam. The indirect effect on the logs sector is due to the increase in production to meet the demand for lumber. The indirect effect on the silviculture sector is due to the raw wood production triggered by the increasing demand for logs. Therefore, the circling of goods spreads greatly upstream. Because it leads to the promotion of forestry, constructing a timber dam has a greater regional economic impact than constructing a concrete dam (Tables 4 and 5). In addition, the large indirect effect on the silviculture sector indicates that the timber dam is a possible supply destination for planted forests of suitable age for logging; finding such a destination is one of the problems of forest management in Japan.

In concrete dam construction, the plywood and glued laminated timber sector accounts for 6% of the indirect effect because of the use of formwork plywood (Table 7). Although this clearly has some regional economic impact on the wood industry, the indirect effect on forestry-related sectors does not appear in the table with a high rank. In other words, for

the promotion of forestry and wood industries, the use of wood for formwork is important, but the use of wood as the main material of the dam body is more so.

In the previous study [19], the economic impacts were compared by unifying only the functional units without considering the appropriate use of the difference in construction costs between timber and concrete dams (Tables 4 and 5). In contrast, this study improved on the previous study and compared the economic impacts by unifying not only the functions, but also the costs (Tables 4 and 10). In the previous study, the comparison of only the economic impact ratio was possible, because the construction costs of timber and concrete dams were different. The difference in cost between the two dams is usually used for other regional promotion measures. Therefore, the previous study was unable to properly compare and evaluate the economic impacts of unifying the costs. In this study, we have addressed this flaw. In particular, the evaluation also includes the economic impact of issuing the difference as premium vouchers by the government. Therefore, the method used in this study is considered to be more appropriate for evaluating the comparison of economic impacts between the construction of timber and concrete dams.

As a result of unifying the functions and costs of the check dams, the total effect of the construction of a timber dam is larger than that of the construction of a concrete dam with issuance of premium vouchers (Tables 4 and 10). However, while the number of sectors with an indirect effect of 10,000 yen or more from timber dam construction was 32, that for concrete dam construction with voucher issuance was larger (55). We assume that this is because the issuance of premium vouchers triggered household consumption and the circling of goods to various industrial sectors: the self-transport, miscellaneous business services, and electricity sectors, which did not rank highly in terms of indirect effect of the construction of a concrete dam, do rank highly in terms of indirect effect of premium-voucher issuance (Tables 7 and 9). Issuing premium vouchers increased distribution, service industries, and electricity consumption in the prefecture at the household level. In addition, although the direct effect of the production value induced by issuing premium vouchers (Table 8) is smaller than that for the construction of a timber dam (Table 4), the number of sectors with an indirect effect of 10,000 yen or more is as large as 43. This proves that issuing premium vouchers circulates goods to various industries in the prefecture. In this study, we focused on the issue of premium vouchers as the regional promotion measure. However, we have to consider the possibility that the results would be different if the difference in construction costs was used for other regional promotion measures. For example, supporting tourism as a regional promotion measure is expected to have an economic impact similar to that of public works projects, as it involves the development of infrastructure for tourism and the construction of accommodation facilities. Therefore, rather than circulating goods to various industries in the region, as is the case with the issuance of premium vouchers, the effect may be limited to fewer industries, as is the case with the construction of timber and concrete dams. In the case of regional promotion measures such as addressing population reduction, the increase in population due to migration increases housing investment and personal consumption, which may expand the economic scale of the region and circulate goods to various industries, as in the case of the issuance of premium vouchers.

Compared to a timber dam, which has a high initial cost, a concrete dam is cheap to construct. Since the difference in costs is returned to the region through the issuance of premium vouchers, concrete dam construction is assumed to be more effective for wide-ranging promotion of the regional economy. However, from the perspective of forestry promotion and conservation, circulating goods to the region through concrete dam construction with a low initial cost and the issuance of premium vouchers is less effective than timber dam construction with a high initial cost, as this study has shown. More generally, in the field of architecture, Adam et al. [20] have reported that mass timber building designs have a high initial cost, but because regionally produced cross-laminated timber (CLT) can be used, the income and employment of regional residents increase when compared with traditional concrete construction methods, which results in a large regional

economic impact. In this study as well, the initial cost of the timber dam is higher than that of the concrete dam; however, because a timber dam can use regional materials, it has a larger total effect and economic impact ratio, in addition to increasing the employment.

It has been shown that the use of regionally produced lumber has a positive impact on the regional economy. We considered the impact of the self-sufficiency ratio of the timber used in the targeted timber dam on the regional economy. The logs and timber sectors are included in the forestry and wood industries used in timber dam construction. The self-sufficiency ratio of these sectors is 100%, because a timber dam is entirely made of regionally produced lumber. However, the actual self-sufficiency ratios of these sectors in Akita Prefecture are 24% for the timber sector and 62% for the logs sector. If we use these self-sufficiency ratios to evaluate the economic impact on the regional economy, the total effect is 45,946,000 yen, which is smaller than the total effect of the construction of the concrete dam and the issue of premium vouchers (46,064,000 yen). In addition, the indirect effect on the timber, logs, and silviculture sectors is also small (timber sector: 1,679,000 yen, logs sector: 541,000 yen, silviculture sector: 611,000 yen). Thus, in the case of a region with low self-sufficiency in the logs and timber sectors, the construction of a concrete dam and the issue of premium vouchers may have a more positive impact on the regional economy than the construction of a timber dam with high initial cost. Therefore, the use of regionally produced timber is also important for the regional economy.

Although the economic structure of Japan is very different from that of Uruguay, the case of the South American country [29] is worth considering. Forestry has been encouraged in Uruguay since the 1960s. The Forestry Law of 1987 introduced subsidies and tax exonerations for the development of forest plantations and wood-manufacturing industries. As a result, forest-sector development has attracted foreign investment, generating income and employment [29]. In this study as well, the total effects on employment and income were larger when production in the forestry and wood industries was induced by timber dam construction than when household consumption in the prefecture was induced through premium-voucher issuance. Thus, the promotion of forestry and wood industries can revitalize the regional economy.

The direct effect of the construction of a concrete dam with premium-voucher issuance on employment (3.42 people) was larger than that of the construction of a timber dam (2.97 people) (Tables 4 and 10). It is assumed that this is because the final demand increase in the latter case was in the construction of a timber dam sector alone, whereas that in the former occurred in multiple sectors owing to the issuance of premium vouchers. By contrast, the indirect effect of the construction of a timber dam on employment was greater than that of the construction of a concrete dam with premium-voucher issuance, indicating that the use of timber may create many jobs in the forestry and wood industries. However, in Japan, the income level of forestry workers is low compared to that of workers in other industries [30], and the incidence of occupational accidents is the highest among all industries [31]. Therefore, although it is important to generate employment in the forestry and wood industries by constructing timber dams, it is also necessary to improve the wages and working environment of forestry workers.

## 5. Conclusions

In this study, we compared the regional economic impacts (unifying both functions and costs) of timber and concrete dam construction in Akita Prefecture, Japan, by extending the latest input–output table. We also considered the regional economic impact when the difference in construction costs was used to encourage growth through the issuance of premium vouchers. Our main findings were as follows:

1.  The total effect of production value induced by constructing a timber dam is 57,706,000 yen (the economic impact ratio is 1.86) and the employment of 5.11 people. When the costs and functions of both types of dams are unified, the regional economic impact and employment effect of the construction of a timber dam are greater than those of the construction of a concrete dam with premium-voucher issuance.

2.  Because concrete dam construction uses plywood in formwork, it causes a ripple effect in the plywood and glued laminated timber sector. On the other hand, timber dam construction causes large ripple effects in the timber sector as well as in the logs and silviculture sectors upstream. Although wood use for formwork is certainly important for the promotion of the forestry and wood industries, wood use for the main materials of the dam body is even more so.

3.  The number of sectors with an indirect effect of 10,000 yen or more is greater for concrete dam construction with premium-voucher issuance than for the construction of a timber dam. Therefore, the construction of a concrete dam and the issuance of premium vouchers are effective in circulating goods to various industries. However, the indirect and total effects, including the effects on the forestry and wood industries, are greater for the construction of a timber dam than for the construction of a concrete dam with the issuance of premium vouchers. Thus, constructing a timber dam is important for the promotion of forestry and wood industries and for the revitalization of the regional economy.

In some timber dams, the structure, tree species, and construction region are different from those of the timber dams examined in this study [32–34]. Therefore, timber dams with different structures, tree species, and regions may have different economic impacts. It is necessary to examine various types of timber dams in future research.

In this study, we proposed a highly versatile method for further investigation. The previous study [19] did not consider the impact of the difference in construction cost between a timber dam and a concrete dam on the regional economy. However, this study proposes a method for appropriately comparing regional economic impacts by unifying the functions and costs of check dams by using the difference in construction costs for regional promotion measures. This method can also be applied to the comparison of the regional economic impacts of structures and public works other than check dams. In addition, the check dams focused on here are public dams. Therefore, this study can play a role in supporting policy making for the use of wood in public works. This is because this study presents methods and data that allow us to scientifically examine the impact of the use of wood in public works on the regional economy. In other words, it will be possible for the government to formulate policies that consider the economic impacts of the use of wood in public works.

**Author Contributions:** Conceptualization: T.H. and C.K.; methodology: T.H., and C.K.; data curation: T.H., and C.K.; writing—original draft preparation: T.H.; writing—review and editing: C.K.; funding acquisition: C.K. All authors have read and agreed to the published version of the manuscript.

**Funding:** This work was supported by the Sumitomo Foundation Fiscal 2019 Grant for Environmental Research Projects (grant number 193259) and JSPS KAKENHI (grant number JP20H04384).

**Institutional Review Board Statement:** Not applicable.

**Informed Consent Statement:** Not applicable.

**Data Availability Statement:** The data are not publicly available due to a confidentiality agreement with the contractors but are available from the corresponding author on reasonable request.

**Acknowledgments:** We want to thank Masahiro Iwaoka of Tokyo University of Agriculture and Technology for the assistance in this study. We thank the contractors and Akita Prefecture for providing invaluable data.

**Conflicts of Interest:** The authors declare no conflict of interest.

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
