# Peer review of "Regional Economic Impacts from Timber Check Dam Construction—A Comparison with Concrete Check Dam Construction, Part II: The Question of Premium Vouchers"

_forests, doi:10.3390/f12030347_

Round 1

Reviewer 1 Report

Regional Economic Impacts from Timber Check Dam Construction—A Comparison with Concrete Check Dam Construction, Part II: The Question of Premium Vouchers

General Comments

This study compares the reginal and economic impacts od the of the construction of timber and concrete check-dam in Akita prefecture, Japan. The manuscript presents several flaws that should be addressed by the authors. Firstly, it is not clear if there is any substantial contribution from this work. This is a case study and the results are not interesting for the international readers. The literature review was not properly addressed and the authors have failed to clearly explain the importance of this work in relation to the current state-of-the-art, as well as to point out which is the substantial contribution of this paper. In general, it is not clear either the relevance, motivation or novelty introduced by their study. An in-depth analysis of the results is missing, and the authors should further develop critical appraisal in their discussion. This paper in the present form is not suitable for publication due to lacks in new findings. In general, the study was poorly developed, and at this point, it does not meet the journal's standards.

Reviewer 2 Report

In the introduction, as well as in the objectives, the Authors should clearly specify the progress made in this manuscript compared to the previous work that they themselves have already published.

Although not particularly novel, the study is interesting mainly on a regional/national scale, since it is an application of good quality.

Finally, I suggest a final English revision to remove some typos.

Round 2

Reviewer 1 Report

Comments to the Author
I have reviewed the manuscript again along with authors responses to my initial comments. The authors have made substantial and valuable changes to the manuscript and I appreciate their efforts to improve the manuscript. I have no further comments.

This manuscript is a resubmission of an earlier submission. The following is a list of the peer review reports and author responses from that submission.